# How Much Backtracking is Enough? Exploring the Interplay of SFT and RL in Enhancing LLM Reasoning

## Abstract

Recent breakthroughs in large language models (LLMs) have markedly advanced their reasoning progress through two broad post-training paradigms: supervised fine-tuning (SFT) and reinforcement learning (RL), particularly on mathematical and logical problems that have verifiable answers. Prior research indicates that RL effectively internalizes search strategies, enabling long chain-of-thought (CoT) reasoning, with backtracking emerging naturally as a learned capability. However, the precise benefits of backtracking—specifically, how significantly it contributes to reasoning improvements and the optimal extent of its use—remain poorly understood. In this work, we systematically investigate the dynamics between SFT and RL on eight reasoning tasks: Countdown, Sudoku, Arc 1D, Geometry, Color Cube Rotation, List Functions, Zebra Puzzles, and Self Reference. Our findings highlight that self-sampled CoT sequences used in SFT as a warm-up do have moderate contribution to RL training, compared with RL without any SFT warm-up; however such contribution diminishes when tasks become increasingly difficult. Motivated by these observation, we introduce a backtracking-centric training recipe. By synthetically varying the number of explicit backtracking steps in the SFT warm-up, we show that (i) longer CoTs containing backtracks stabilize and amplify RL, and (ii) the optimal backtrack depth scales with task difficulty—zero for Arc 1D, one for Countdown, and five for Sudoku—yielding up to a 28.9% absolute accuracy boost at the 3B parameter scale. Collectively, our controlled experiments provide concrete guidance for constructing training mixtures that reliably push LLM reasoning beyond current boundaries.

## 1 Introduction

Recent large language models such as DeepSeek-R1 (DeepSeek-AI et al., 2025) and OpenAI (2024)'s O1 have demonstrated remarkable reasoning abilities especially on various complex reasoning tasks. Their success stems from allocating substantially more inference-time compute, letting the model perform an internal search via extended chains of thought before committing to an answer (Snell et al., 2024). Reinforcement learning has emerged as the most effective way to unlock these long reasoning traces (Shao et al., 2024; Team et al., 2025; Schulman et al., 2017; Zelikman et al., 2022; Rafailov et al., 2024). What is new in the DeepSeek-R1 era is the shift from open-ended "process" rewards—which score intermediate reasoning steps (Lightman et al., 2023; Guan et al., 2025; Zhang et al., 2025; Zheng et al., 2024)—to verifiable, rule-based

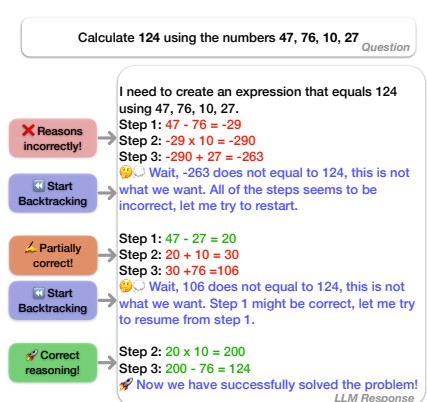

Figure 1: LLM post-trained with synthetic backtracking is able to backtrack incorrect steps.

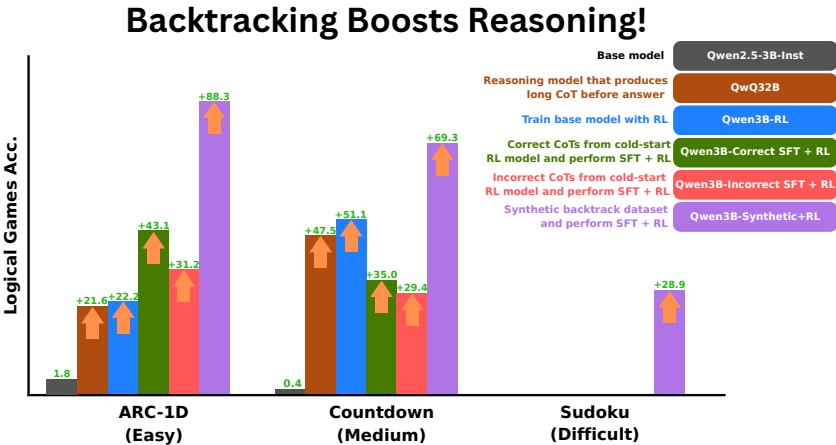

Figure 2: We engineer SFT data with synthetic backtracks to achieve significant improvements over three reasoning tasks, Arc, Countdown, and Sudoku, each representing a different difficulty.

reward functions that directly validate final answers against symbolic checkers or ground-truth programs (Zeng et al., 2025; Chen et al., 2025). These programmatic rewards give the agent a sharp learning signal, guiding exploration toward solution trajectories that yield correct outputs and thereby enabling the consistently deep chains of thought seen in today's best reasoning models.

Although reinforcement-learning often strengthens an LLM's reasoning, the scope and limits of these gains remain under-explored. Gandhi et al. (2025) document this uncertainty by comparing two closely matched model families. RL produces a dramatic jump for Qwen, yet yields only marginal gains for Llama. One hypothesis credits Qwen's advantage to its frequent use of thinking tokens such as "wait" and "verify" (Xie et al., 2025). Gandhi et al. (2025) test this by first supervised-finetuning Llama on chain-of-thought data and then applying RL; the CoT warm-up induces richer cognitive behaviors such as backtracking or self-checking and yields larger RL gains than training vanilla Llama directly. These and related studies suggest that RL acts more as an amplifier on different behavioral patterns, such as backtracking that's present in the base model. Some evidence points to RL leveraging patterns already innate in pre-training data (Zhao et al., 2025) and generalizing them to novel tasks (Chu et al., 2025). Conversely, other work shows that simply generating longer reasoning traces does not guarantee better accuracy (Qin et al., 2025; Wang et al., 2025; Yue et al., 2025).

Regardless, the ability to backtrack, which allows models to rethink and revise its previous solutions, are commonly observed across the strongest reasoning models (DeepSeek-AI et al., 2025; OpenAI, 2024). Backtracking is not merely a specific heuristic but a universal operator for tasks formulated as tree search—a structure essential for exploration, planning, and multi-hop reasoning (Qin et al., 2025). Unlike domain-specific heuristics, backtracking provides a general-purpose interface for navigating non-linear solution spaces. In practice, backtracking implicitly includes both multiple verification strategies and alternative solution exploration. For a model to decide when to backtrack, it must first verify or reflect its current partial solution. Verification or reflection provide the signal that a current path is incorrect, but backtracking is the necessary action to recover from that state. Because of this, we aim to provide a principled understanding on the training interactions between SFT and RL through warm-up data perspective on backtracking frequency. In this work, we answer several questions: is warm-up stage necessary for RL? What kinds of SFT warm-up matter to RL? How many backtracking steps are optimal, and when do they become counter-productive?

To rigorously answer the questions above, we design a suite of controlled experiments. We systematically compare pure RL (RL on base model without any reasoning specific SFT warm-up) and cold start RL with self-sampled CoTs as warm-up. Our results show that, while pure RL already yields some improvements, even simple SFT warm-ups—such as self-sampling from pure RL'ed model and re-performing SFT+RL pipeline (we refer to this as cold-start RL)—can further enhance per-

Table 1: A brief explanation of each SFT warm-up setup explored in this paper.

| Abbr. | Explanation |
| --- | --- |
| No-SFT | RL with no SFT (Pure RL) |
| Self-sampled SFT | SFT on model's own generated trajectories |
| Synthetic backtrack SFT | SFT on synthetic trajectories with explicit backtracking |
| Shuffled SFT | SFT on trajectories being shuffled with other problem's trajectories |

formance. For a subset of three tasks, we curate synthetic datasets using depth-first search (DFS) or heuristic search, injecting varying numbers of synthetic backtracks into the SFT warm-up. We find that backtracking demonstrably benefits RL training, with more challenging tasks requiring deeper backtracking to maximize gains.

Taken together, our controlled experiments reveal a precise recipe for effective RL: initializing with synthetic SFT data containing an appropriate number of backtracks—matched to the difficulty of the task—consistently pushes the model beyond its base reasoning capabilities.

Our research provides the following key contributions:

- In Section 3, we conduct controlled investigation on eight reasoning tasks and find that short self-sampled data that rarely exhibit any behavioral patterns such as "wait" do have additional contribution to the RL training compared to pure RL, contrary to prior findings.(Gandhi et al., 2025).
- In Section 4, we propose a backtrack-centric data and training recipe to allow base model scale efficiently to combinatorially hard tasks with RL.
- In Section 5, we find that RL post-training initialized from even incorrect CoTs can improve performance.

## 2 METHODOLOGY

In this study, we conduct a comprehensive analysis using controlled experiments to investigate the interplay between SFT and RL, focusing on how different training data mixtures influence RL outcomes. Specifically, we examine which types of SFT warm-ups benefit RL and seek to understand the underlying reasons. Table 1 presents the SFT setups we evaluate: no SFT, self-sampled SFT, synthetic backtracking SFT, and shuffled SFT. Ultimately, we aim to equip the model with the ability to tackle difficult problems beyond its base capabilities.

We employ a total of 8 reasoning tasks, adopted from Reasoning Gym (Thought, 2025), and select subsets to further curate synthetic to ensure our experiments provide analytical insights.

### 2.1 REASONING GAMES

**Countdown** The goal is to construct an arithmetic expression using a set of numbers to reach a specific target value. Each number can be used at most once, and operations are limited to basic arithmetic: addition, subtraction, multiplication, and division.

**Sudoku** The objective is to fill a $9 \times 9$ grid so that each row, column, and $3 \times 3$ subgrid contains all the digits from 1 to 9 exactly once.

**Arc 1D** The objective is to learn and apply a transformation rule that maps a one-dimensional input grid to a corresponding output grid, given several input–output examples.

**Advanced Geometry** Advanced Geometry contains three sub-tasks. In the **angle measurement** sub-task, the goal is to compute the internal angle at a specific vertex of a triangle, given the input of three vertex coordinates in the Cartesian plane. In the **orthocenter** sub-task, the goal is to determine the orthocenter of a triangle—the point where all three altitudes intersect, given the input of the

triangle's three vertex coordinates. In the **incircle radius** sub-task, the goal is to compute the radius of the incircle of a triangle, given the input of the coordinates of its three vertices.

**Color Cube Rotation**   In the game of Color Cube Rotation, the goal is to determine the color of a specific side of a cube after a sequence of 3D rotations, given the input of its initial face-color configuration.

**List Functions**   The goal is to generate an output list by identifying and applying an implicit rule that maps each input list to its corresponding output, given examples.

**Zebra Puzzles**   The goal is to determine a specific attribute—such as the name of the person in a given house—based on a set of logical constraints, given the input of several people and their distinct characteristics (e.g., favorite drink, pet, or phone).

**Self Reference**   The goal is to determine how many consistent truth assignments exist for a set of self-referential statements, given the input of seven logically interconnected claims. Each statement makes assertions about the truth or falsity of other statements (or the total number of true/false ones).

## 2.2 MODEL AND TRAINING

We use Qwen2.5 family of models (Qwen et al., 2025), primarily focusing on Qwen2.5-3B-Instruct for supervised-finetuning and reinforcement learning training (Pan et al., 2025; Sheng et al., 2024). We also include Qwen2.5-7B-Instruct for baseline comparison. The evaluation metrics is pass@1 where we only focus on the correctness of the final answer, which should be placed inside `<answer></answer>` tokens.

For both SFT and RL training, we adopted code from Sheng et al. (2024); Pan et al. (2025), the specific rollout lengths depend on whether the model is initialized from long or short CoT. Typically, for RL training initialized from long CoTs, we use rollout length from 4k to 8k tokens, depending on the tasks. For short CoTs, we use rollout length from 1k to 2k tokens, depending on the tasks (Appendix B).

We adopt rule-based rewards (DeepSeek-AI et al., 2025), where format accuracy (successfully generate a pair of `<think></think>` and `<answer></answer>` tokens with thinks tokens come before answer tokens) is rewarded 0.1 point, and the answer accuracy-exact match with the ground truth-is rewarded 0.9 point (Appendix C). Together, a problem may be rewarded a maximum of 1.0 point and the minimal of 0.0 point.

## 3 MAXIMIZING BASE MODEL REASONING CAPABILITIES

To systematically investigate how SFT interacts with reinforcement learning in enhancing large language models' reasoning capabilities, we scientifically address a foundational question: Is a warm-up stage necessary for effective RL? Specifically, we begin with two distinct approaches—pure RL (no SFT warm-up) and cold-start RL with a short self-sampled CoT warm-up—to establish baselines and evaluate their efficacy across diverse reasoning tasks. These experiments demonstrate that, although pure RL is already effective, incorporating a simple self-sampled CoT warm-up further improves performance. However, both methods face significant limitations when applied to more challenging reasoning tasks, such as Sudoku.

### 3.1 PURE RL - WORKS ALBEIT LIMITED

We first explored pure RL, a direct RL approach without any reasoning specific SFT warm-up. The motivation behind this was to establish a clear baseline and determine whether the model could independently internalize effective search strategies and concise reasoning through pure RL exposure. In Figure 3, we observe just pure RL itself is able to witness improvements in model's reasoning abilities, compared to their base model. We manually inspect a few of the generated trajectories and find that pure RL enables the model to internalize rudimentary strategies, such as explicit verbalized reasoning (e.g., in Appendix D.2) and concise thinking (e.g., in Appendix D.1). It is noticeable that

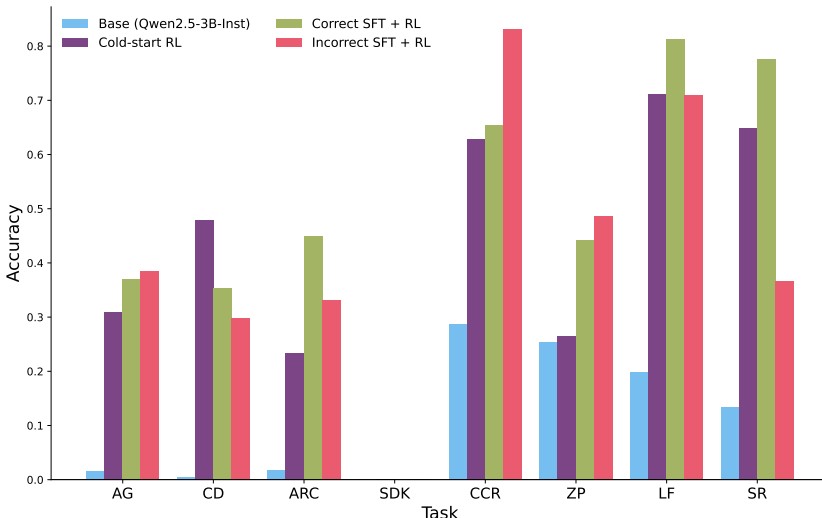

Figure 3: In domain evaluation comparison bewteen base model, pure RL, correct self-sampled SFT + RL, and incorrect self-sampled SFT + RL. AG means Advanced Geometry, CD means Countdown, ARC means Arc 1D, SDK mean Sudoku, CCR means Color Cube Rotation, ZP means Zebra Puzzles, LF means List Functions, SR means Self Reference

for most tasks, pure RL naturally produces linear trajectories without backtracking. For Countdown, the model produces backtracking but it is unstructured. Due to probable pre-training data exposure, the model occasionally attempts to correct itself. However, these corrections are stochastic and often "lazy"—retrying the whole calculation rather than pinpointing the error.

Additionally, training the model in one environment also improves performance in other tasks as shown in Appendix H. Training in Advanced Geometry environment induces good generalization to Zebra Puzzles and List Functions; training in Countdown induces generalization to Advanced Geometry. However, limitations quickly became apparent, particularly on combinatorial problems like Sudoku and Zebra Puzzles, indicating that pure RL alone is insufficient for complex reasoning tasks.

## 3.2 Self-sampled short SFT as warm-up goes beyond pure RL

Building on the insights from pure RL, we hypothesized that a supervised fine-tuning warm-up using short self-sampled CoTs could further enhance reasoning performance by priming models with initial cognitive strategies.

**Training and evaluating setup**  In this series of experiments, we first perform specialized pure RL training starting from Qwen2.5-3B-Instruct (Qwen et al., 2025) across 8 reasoning datasets (Thought, 2025). After obtaining 8 RL'ed models, we subsequently collect their reasoning trajectories on the same tasks they are finetuned on, but with problems generated from a different seed (different problems). We purposefully separate the trajectories that lead to correct answer and those that lead to incorrect answers, from which we further perform the complete SFT + RL post-training pipeline starting again from vanilla Qwen2.5-3B-Instruct, and obtain 15 specialized models (7 RL'ed models initialized from correct SFT, and the other 8 initialized from incorrect SFT). We maintain identical gradient steps between pure RL and self-sampled SFT + RL to ensure fair comparison. A comprehensive evaluation is subsequently performed for all 23 models (three models per task with the exception of Sudoku, which we discuss below) in Figure 3.

**Self-sampled CoTs are effective warm-ups**  Short self-sampled CoT warm-up consistently demonstrated performance gains across nearly all tasks compared to pure RL in Figure 3. Specifically, we observed significant performance improvements on tasks such as ARC and List Functions.

It is also worth noting that even incorrect trajectories warm-up can improve downstream RL performance on some of the tasks. We will discuss more details in later section.

Notably, tasks like Sudoku still posed substantial challenges, highlighting the limitations of self-sampled CoTs for deeper, combinatorial reasoning. In fact, the Cold-Start RL pipeline effectively breaks down for Sudoku because the model cannot self-generate the necessary correct SFT initialization data, which illustrate a fundamental brittleness of the Cold-Start RL paradigm. These results underscore the incremental yet limited nature of cold-start RL, motivating further exploration of more sophisticated strategies.

# 4 PUSHING BEYOND BOUNDARIES WITH BACKTRACKING

Recognizing the limitations identified in previous methods, particularly the failure of both cold-start RL and short-CoT warm-ups on combinatorially hard tasks such as Sudoku, we propose a backtracking-centric training paradigm to push models beyond their baseline reasoning limits. Empirical analyses of the strongest reasoning models reveal spontaneous use of backtracking (e.g., self-correct, multiple attempts), suggesting that explicit exposure to backtracking traces during SFT can seed deeper search policies that subsequent RL can refine. We therefore hypothesize that integrating controlled amounts of backtracking into the SFT stage will enable models to internalize more powerful search strategies and unlock substantial gains on challenging tasks.

**Building synthetic backtracking datasets**   To systematically evaluate the impact of backtracking, we selected three representative reasoning tasks-Countdown, Arc 1D, and Sudoku-to construct synthetic datasets with varying numbers of backtrack, as well as optimal trajectories, and study whether, and how RL amplifies such behavior. The three tasks are selected because they are the more challenging subset based on vanilla model's baseline performance in Figure 3.

To construct synthetic datasets for Countdown and Sudoku, we use DFS solver and create a tree structure where each node represents an intermediate step (e.g., an operation between two numbers for Countdown; fill one grid with a valid number for Sudoku). Then we start from the solution node back to the root node to build optimal solutions, and include incorrect branches as detours and backtracks. For Arc 1D, we design a heuristic search that applies hand-crafted transformation functions with step-wise, verbalized reasoning. A detour or backtrack is created by choosing an incorrect transformation and then retrying.

**Quantifying the Impact of Backtracking in SFT Warm-up**   Our controlled experiments reveal that the exposure to backtrack systematically improves RL training efficiency as shown in Figure 4. However, the number of backtracks needed depends on the difficulty of the problem:

1. **Countdown — Moderate.** Every puzzle provides 4–6 numbers and a target value, producing a medium-sized arithmetic search tree. Although the Qwen2.5-3B-Instruct can solve 0.4% of the problems. it can generate plausible attempts. We found that using exactly **one** backtrack is most optimal.

2. **Arc 1D — Moderate/Easy.**   Tasks are solved by applying a small pool of grid-transformation heuristics; the same baseline attains 1.8% accuracy. Using **zero** backtrack is optimal.

3. **Sudoku — Hard.** Each board contains 30–60 empty cells; every cell admits multiple candidates, yielding an astronomically large search space. A vanilla The Qwen2.5-3B-Instruct baseline solves none of the test instances. Using **five** backtracks is optimal.

Our controlled experiments confirm a consistent pattern: harder combinatorial problems require deeper back-tracking traces to seed RL, whereas easier tasks are best served by shallow or even optimal demonstrations.

For **Countdown**, RL initialized with one backtrack achieves the highest reward curve (Figure 4b), whereas RL from zero backtrack performs substantially worse. This suggests that one backtrack initialization enables the model to internalize and execute efficient search strategies. Evaluation results confirm this, with the one-backtrack model attaining 69.7% accuracy (Figure 4d), outper-

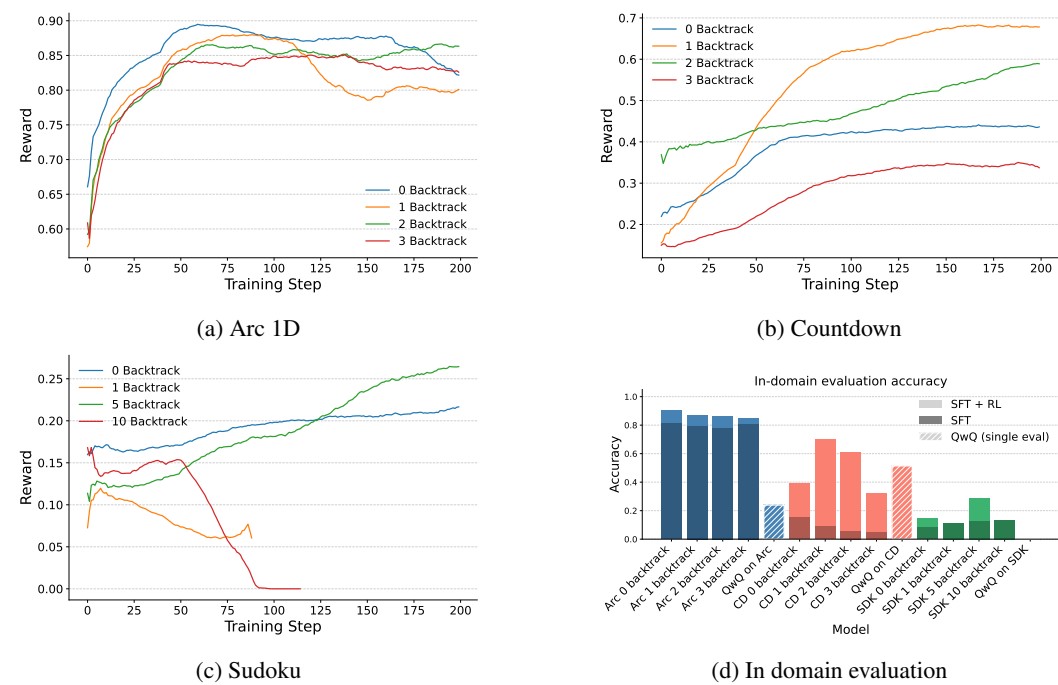

(a) Arc 1D

(b) Countdown

(c) Sudoku

(d) In domain evaluation

Figure 4: RL training reward trajectories for (a) Arc 1D, (b) Countdown, and (c) Sudoku, as well as (d) each model's in domain evaluation comparison between the SFT and SFT + RL training paradigm.

forming QwQ-32B (Qwen et al., 2025) at $51.5\%$ (8K context window), while the zero-backtrack model achieves only $38.9\%$.

For **Sudoku**, initializing PPO with five backtracks is necessary for stable, effective training. Too few or too many backtracks (e.g., one or ten) lead to model degeneration. Notably, pure RL and cold-start RL on self-sampled short CoTs fail to train on Sudoku, with reward trajectories stagnating at 0.1. Backtracking consistently outperforms optimal traces, as shown by $28.9\%$ accuracy for the five-backtrack model versus $14.4\%$ for the zero-backtrack model. This also surpasses the QwQ-32B baseline, which achieves $0.0\%$ (Table G).

For **Arc 1D**, the easier task among three, performance declines as the number of backtracks increases. Models trained with zero backtracks consistently outperform others, with the zero-backtrack achieving $90.8\%$ accuracy, significantly exceeding QwQ-32B's $24.0\%$.

These results clearly illustrate that optimal backtracking scales with task difficulty, highlighting its crucial role in enabling deep, effective reasoning.

## 5 ADDITIONAL ANALYSIS OF INTERPLAY BETWEEN SFT AND RL

In this section, we document additional findings and include analysis throughout our goal of breaking the base boundry of reasoning.

### 5.1 INCORRECT TRAJECTORIES ARE STILL USEFUL

Models initialized from incorrect reasoning CoTs display similar behaviors with those from correct trajectories. Figure 3 shows that incorrect SFT models post RL see the same trend of performance increase or decrease with correct SFT models on in-domain evaluations, with the only exception being self reference task. While initially counter-intuitive, this finding aligns with recent evidence that RL can quickly overwrite semantic errors provided the demonstrations respect the expected CoT

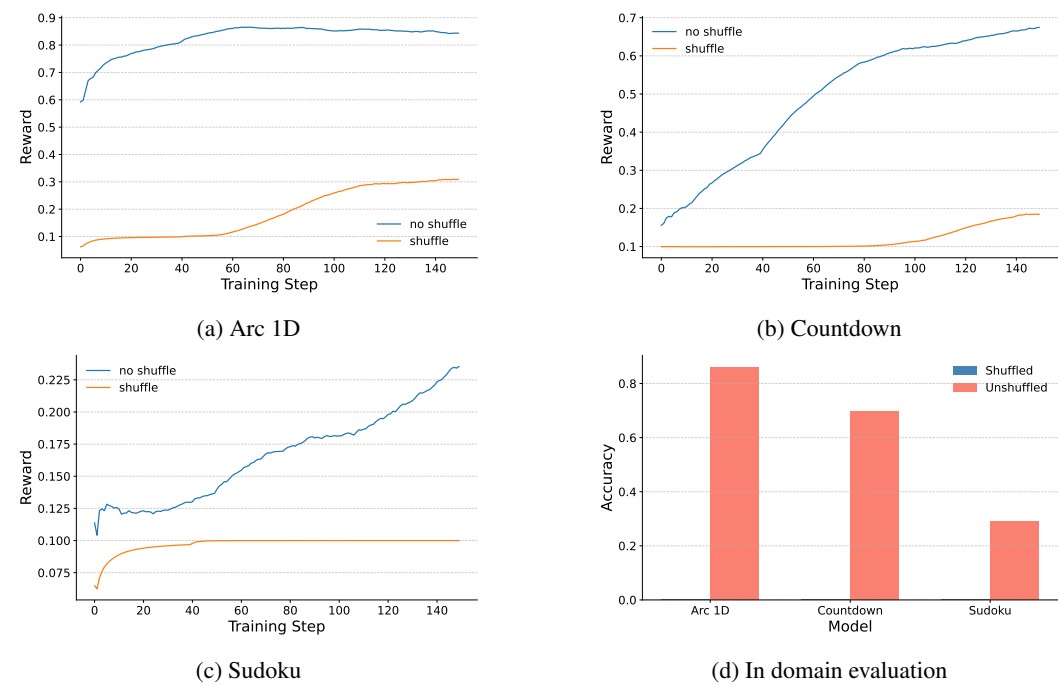

Figure 5: RL training reward trajectories for models post-trained on shuffled and not shuffled synthetic dataset of (a) Arc 1D, (b) Countdown, and (c) Sudoku, as well as (d) each model's in domain evaluation comparison. Models with initialization from shuffled datasets achieve $0.0\%$ in accuracy.

format (Setlur et al., 2024). We conjecture that syntactic scaffolding alone primes the base model with an executable search blueprint that subsequent RL can refine.

### 5.2 RL IS SENSITIVE TO SHUFFLED SFT

To probe the limits of semantic noise that an SFT warm-up can tolerate, we deliberately broke the alignment between prompts and completions: within the best-performing backtracking dataset in Section 4. we randomly shuffled questions and their corresponding CoT-answer pairs. This manipulation preserves syntactic structure—every example still contains well-formatted trajectories while obliterating the underlying reasoning. From Figure 5, we see that the training becomes extremely ineffective, suggesting that despite having the right reasoning structure, RL training is easily vulnerable to the internal inconsistency between answer and completion during SFT. In fact, we find that, for Countdown and Arc 1D in Figure 5b and 5a, even when exposure is capped at fewer than 5k, models recover only a negligible reward signal after roughly 60–80 PPO updates before stagnating. These results contrast sharply with the resilience to incorrect but aligned CoTs observed in previous subsection, indicating that RL is highly sensitive to internal inconsistencies between a reasoning trace and its stated answer. Manual inspection of generated outputs (Figure D.3) reveals that the policy over-fits to spurious prompt–completion pairings, entangling incorrect intermediate states that PPO cannot unlearn.

## 6 RELATED WORK

### 6.1 TEST TIME SCALING

Recent work has shown scaling up language model output length during test time can be more effective than scaling pretraining (Snell et al., 2024; Wang et al., 2024; Jiang et al., 2023; Saad-Falcon et al., 2024). As language models become increasingly larger and pre-train on more and more enormous corpus; this third axis of scaling offers a promising gains in performance while remaining budget friendly (Wang et al., 2025). There are mainly two directions for scaling up:

parallel sampling and sequential search. Common parallel sampling methods such as best-of-n rely on LLMs to propose answers independently N times (Li et al., 2024); whereas sequential search generates each response in sequence conditioned on previous attempt (Qu et al., 2024; Feng et al., 2024; Gandhi et al., 2024; Yang et al., 2025). There have also been attempts to combine the two and incorporate external verifiers through algorithms like Monte-Carlo Tree Search (Guan et al., 2025; Feng et al., 2024). Our work extends this line of research by examining the optimal training mixtures for efficiently scaling up model generation for reasoning.

### 6.2 REINFORCEMENT LEARNING FOR REASONING

A novel dimension of scaling inference-time compute for language models has recently gained prominence through the emergence of specialized "reasoning models" trained via reinforcement learning (RL). Prior works predominantly explored on-policy and off-policy RL methods (Zelikman et al., 2024; 2022; Kazemnejad et al., 2024; Phan et al., 2023) and saw moderate successes. However, recent approaches such as DeepSeek-R1 (DeepSeek-AI et al., 2025) have revitalized interest in RL training utilizing verifiable rewards (RLVR), driven by Proximal Policy Optimization (PPO) and its memory-efficient variant, Group Relative Policy Optimization (GRPO) (Shao et al., 2024). A key driver behind RLVR is the emphasis on rule-based rewards derived directly from final outputs, replacing traditional reward models. This new RL training paradigm has notably revealed critical "aha moments," where models spontaneously generate distinctive tokens such as "wait," indicative of active internalization of search strategies and optimized trajectory discovery.

One prevailing hypothesis is that such fine-tuning allows language models to internalize search algorithms during inference-time explicitly. Methods such as Stream of Search (Gandhi et al., 2024), for instance, fine-tune models using linearized search trajectories, empowering continuous and coherent search capability within single-output generation (Yang et al., 2025; Lehnert et al., 2024). Alternatively, another hypothesis suggests that RL training utilizing verifiable rewards does not fundamentally induce novel reasoning skills but rather exploits and amplifies capabilities already learned during the pretraining phase (Yue et al., 2025). This claim is supported by observations that base pretrained models can achieve comparable or superior performance in metrics such as pass@k compared to their RL-trained counterparts. Nevertheless, the precise mechanisms behind these emergent reasoning capabilities remain poorly understood. Our research addresses this gap by systematically investigating how RL leverages SFT and the associated data mixture to unlock the reasoning capacities.

There are a few concurrent efforts investigating how fine-grained behavioral signals influence RL-trained reasoning models. Gandhi et al. (2025) analyzes four behavioral cues, demonstrating that such cues can accelerate RL convergence. We perform controlled experiments on more tasks and study various factors of SFT mixtures. Qin et al. (2025) examines whether explicit backtracking traces improve performance across CountDown and Sudoku. We go further by rigorously controlling trace correctness, format, and synthetic backtrack frequency. Finally, Zhao et al. (2025) explores how variations in pre-training corpus composition interact with subsequent post-training, whereas we emphasize on the post-training affects the reasoning ability.

## 7 CONCLUSION

In this work, we thoroughly inspect the interplay between SFT and RL for reasoning tasks through controlled experiments, emphasizing on the contribution of training data mixtures towards effective RL post-training. We found that Qwen2.5-3B-Instruct models, when performing pure RL, demonstrates two distinct reasoning pattern: verbalized searching and backtracking that iteratively tries to find the correct answer, and concise thinking which solves the problem in one go. To solve combinatorially hard problems like Sudoku, we incorporate backtracking into our training paradigm to achieve around 29% improvement. Further, we have empirical evidence demonstrating that, when initializing RL training from models that have seen incorrect demonstrations, training trajectories are still stable, only slightly shy of the performance of models initialized from correct CoT data. Leveraging the characteristics of pretrain and SFT data to continually scale model performance through RL remains an intriguing future direction.

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

## A USAGE OF LLM

We utilize LLM to assist some of the paper, specifically polish grammar and sentences. We also utilized LLMs for simple code development, for example writing a initial evaluation code. Lastly, we also use LLM to write the code for figure generation.

## B MODEL TRAINING DETAILS

For RL, we investigate both PPO and GRPO. We use learning rate of $1e-6$ for both actor and critic, KL control coefficient of 0 (no KL penalty), gradient clipping of 1.0, entropy coefficient of 0.001, value coefficient of 1.0, gae lambda of 1.0, gamma of 1.0. These hyperparamters are held constant across all runs, with the only difference being the response length to accommodate for problems with longer solution trajectories such as Sudoku. All experiments are done on 4 A100s.

## C REWARD SETUP

**Countdown** We expect solution to not include the equal sign

**Advanced Geometry** For angle measure task: we expect a solution string rounded to two decimal places with degrees symbols attached. Failure to include degree symbol, or not rounding to two decimal places will be deemed incorrect. For orthocenter task: we expect a solution to give the coordinate either in the form "(1.000 2.000)" or "(1.000, 2.000)" or "(1.000,2.000)". Failure to include "()" or not rounding to 3 decimal places will be deemed incorrect. For incircular radius task: we expect a solution string in the format of "1.123". Failure to round to 3 decimal places will be deemed incorrect.

**Arc 1D** We expect a solution string in the format of "1 2 3", other formats such as "1, 2, 3" will be deemed incorrect.

**Sudoku** We expect a solution string in the format of "1 2 3\n4 5 6\n7 8 9", other formats such as "1, 2, 3\n4, 5, 6\n7, 8, 9\n" will be deemed incorrect and only receive format score. Parsing includes generating a list of lists from the solution string.

**Color Cube Rotation** The solution parser is case insensitive. We expects a solution string in the format of "cyan" or "Cyan". Parsing includes lower case all letter.

**Zebra Puzzles** We expect a solution string in the format of "bob" or "Bob". Parsing includes lower case all letters.

**List Functions** We expect a solution string either in the format of "[1 2 3]" or "[1, 2, 3]" or "[1,2,3]". Failure to include "[]" will be deemed incorrect. Will parse to a numpy array.

**Self Reference** We expect a solution in the format of a single digit like "4". Parsing includes converting the solution string to an integer.

## D EXAMPLE MODEL OUTPUTS

### D.1 LIST FUNCTIONS PURE RL MODEL OUTPUTS

```
<|im_start|>assistant
Let me solve this step by step.
<think> The rule in the examples seems to be reversing the order
of the elements in the list. Let's apply this rule to the input
list [74, 72, 49, 43, 41, 3, 30, 85, 48]. Reversing [74, 72, 49,
43, 41, 3, 30, 85, 48] gives us [48, 85, 30, 3, 41, 43, 49, 72,
74]. </think>
```

```
<answer> [48, 85, 30, 3, 41, 43, 49, 72, 74] </answer><|im_end|>
```

## D.2 COUNTDOWN PURE RL MODEL OUTPUTS

```
<|im_start|>assistant
Let me solve this step by step.
<think> First, I need to create an expression that equals 194
using the numbers 14, 17, 2, 45, and 8. Let's try some
combinations. If I add 45 and 17, I get 62. Then if I add 14 to
62, I get 76. Adding 8 to 76 gives me 84. If I add 2 to 84, I
get 86. I need to reach 194, so I need to try different
combinations. Let's try 45 * 2 = 90, then 90 + 17 = 107, then
107 + 14 = 121, then 121 + 8 = 129. This doesn't work. I need
to find a better combination. Let's try 17 * 8 = 136, then 136
+ 45 = 181, then 181 + 14 = 195, then 195 - 2 = 193. Still not
194. I need to find a different approach. Let's try 2 * 14 =
28, then 28 + 45 = 73, then 73 + 17 = 90, then 90 + 8 = 98.
Still not 194. I need to find a different combination. Let's
try 45 * 2 = 90, then 90 + 14 = 104, then 104 + 17 = 121, then
121 + 8 = 129. Still not 194. I need to find a different
approach. Let's try 17 * 14 = 238, then 238 - 45 = 193,
then 193 + 8 = 201, then 201 - 2 = 199. Still not 194. I need
to find a different combination. Let's try 14 * 2 = 28, then
28 + 45 = 73, then 73 + 17 = 90, then 90 + 8 = 98. Still not
194. I need to find a different approach. </think>
```

## D.3 COUNTDOWN RL MODEL FROM SHUFFLED SYNTHETIC DATA OUTPUT

```
<|im_start|>assistant
Let me solve this step by step.
<think>
Step 1: 51 - 23 = 28. Step 2: 28 * 36 = 1008. Step 3: 1008 -
57 = 951. Step 4: 951 - 48 = 885. Wait, this doesn't lead to
the correct solution. 885 is not the correct answer. Let me
go back to step 2 and keep thinking from there.Step 3: 1008
- 27 = 981. Step 4: 981 - 48 = 933. This matches the problem
statement. This is the solution.
</think>

<answer>51 - 36 * 36 - 57 - 48 - 27</answer><|im_end|>
```

# E QWEN2.5-3B-INSTRUCT TRAINING TOKEN COUNTS

Table 2: Token counts for different datasets and training methods on Countdown.

| Dataset / Method | # Tokens |
|---|---|
| Optimal synthetic dataset | 108.81 |
| 1 backtrack dataset | 796.16 |
| 2 backtrack dataset | 883.26 |
| 3 backtrack dataset | 745.31 |
| Correct cold start RL | 625.03 |
| Incorrect cold start RL | 221.37 |
| Pure RL | 442.40 |

Table 3: Token counts for different datasets and training methods on Arc 1D.

| Dataset / Method | # Tokens |
|---|---|
| Optimal synthetic dataset | 270.43 |
| 1 backtrack dataset | 473.64 |
| 2 backtrack dataset | 664.81 |
| 3 backtrack dataset | 889.47 |
| Correct cold start RL | 127.70 |
| Incorrect cold start RL | 352.76 |
| Pure RL | 118.80 |

Table 4: Token counts for different datasets and training methods on Sudoku.

| Dataset / Method | # Tokens |
|---|---|
| Optimal synthetic dataset | 4702.90 |
| 1 backtrack dataset | 7028.17 |
| 5 backtrack dataset | 7690.53 |
| 10 backtrack dataset | 7607.92 |
| Correct cold start RL | |
| Incorrect cold start RL | |
| Pure RL | 449.55 |

## F  LLAMA3.2-3B-INSTRUCT RESULTS

Table 5: In domain evaluation accuracy of Llama3.2-3B-Instruct trained on Arc 1D with varying backtracking levels.

| Data | Evaluation Acc (%) |
|---|---|
| **0 backtrack** | **91.5** |
| 1 backtrack | 85.3 |
| 2 backtrack | 87.2 |
| 3 backtrack | 82.6 |

Table 6: In domain evaluation accuracy of Llama3.2-3B-Instruct trained on Countdown with varying backtracking levels.

| Data | Evaluation Acc (%) |
|---|---|
| 0 backtrack | 37.6 |
| **1 backtrack** | **63.0** |
| 2 backtrack | 48.2 |
| 3 backtrack | 44.4 |

Table 7: In domain evaluation accuracy of Llama3.2-3B-Instruct trained on Sudoku with varying backtracking levels.

| Data | Evaluation Acc (%) |
|------|--------------------|
| 0 backtrack | 9.6 |
| 1 backtrack | 23.4 |
| 5 backtrack | 25.2 |
| **10 backtrack** | **26.9** |

## G QwQ-32B EVALUATION RESULTS

Table 8: Baseline accuracy of vanilla QwQ-32B on 6 different reasoning tasks.

| Model | AG | CD | ARC | SDK | CCR | LF |
|-------|-----|-----|-----|-----|-----|-----|
| QwQ-32B | 0.344 | 0.515 | 0.240 | 0.000 | 0.135 | 0.748 |

## H SELF SAMPLED CoTs MODEL EVALUATIONS

Table 9: Evaluation of Qwen2.5-3B-Instruct baseline and their pure RL'ed models. The row names are the model names (which task they have been RL'ed on), and the column names represent the evaluation tasks.

| Model | AG | CD | ARC | SDK | CCR | ZP | LF | SR |
|-------|-----|-----|-----|-----|-----|-----|-----|-----|
| Qwen2.5-3B-Instruct | 0.015 | 0.004 | 0.018 | 0.000 | 0.286 | 0.254 | 0.199 | 0.134 |
| Qwen2.5-7B-Instruct | 0.052 | 0.019 | 0.064 | 0.000 | 0.281 | 0.388 | 0.314 | 0.138 |
| AdvGeom | 0.309 | 0.019 | 0.021 | 0.000 | 0.244 | 0.319 | 0.241 | 0.131 |
| Countdown | 0.043 | 0.479 | 0.033 | 0.000 | 0.221 | 0.143 | 0.214 | 0.086 |
| Arc1D | 0.001 | 0.010 | 0.234 | 0.000 | 0.241 | 0.319 | 0.299 | 0.156 |
| Sudoku | 0.012 | 0.002 | 0.025 | 0.000 | 0.259 | 0.269 | 0.217 | 0.148 |
| ColorCube | 0.149 | 0.019 | 0.030 | 0.000 | 0.629 | 0.331 | 0.247 | 0.105 |
| Zebra | 0.076 | 0.019 | 0.024 | 0.000 | 0.346 | 0.265 | 0.240 | 0.112 |
| ListFunc | 0.120 | 0.015 | 0.028 | 0.000 | 0.340 | 0.308 | 0.712 | 0.125 |
| SelfRef | 0.000 | 0.007 | 0.054 | 0.000 | 0.268 | 0.352 | 0.185 | 0.648 |

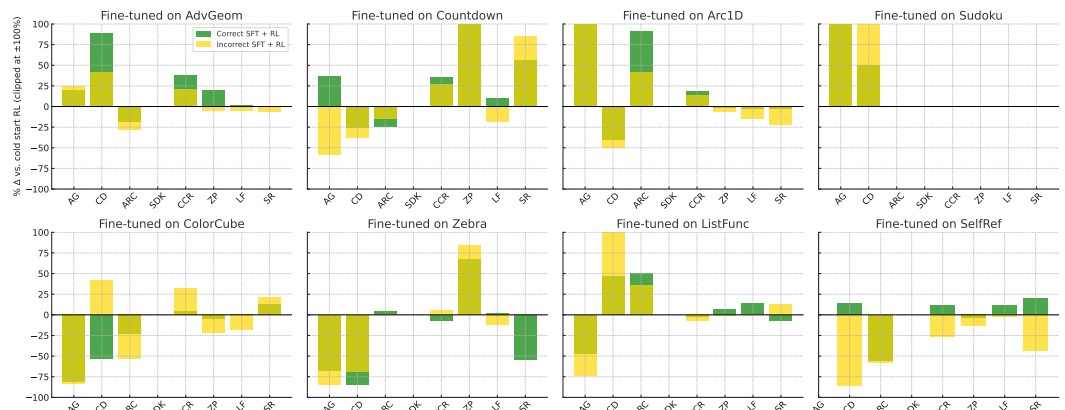

Figure 6: Correct and incorrect short CoTs RL model evaluation

# I    RESPONSE LENGTH COMPARISON

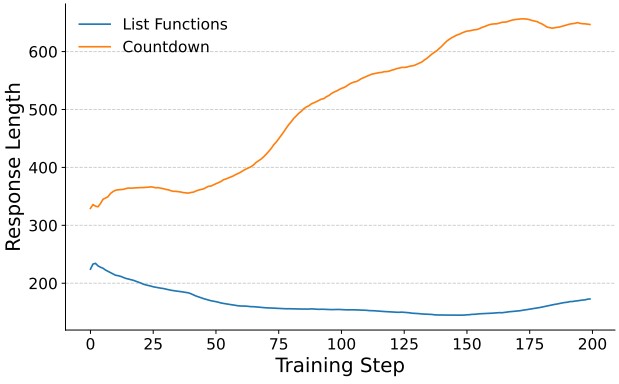

Figure 7: Response length comparison between List Functions and Countdown

# J    LICENSES

All models used (qwen models) are under Apache License Version 2.0. All assets released by this work can be freely used.

All datasets used (reasoning gym) are under Apache License Version 2.0. All assets released by this work can be freely used.

