# OpenReview forum: "How Much Backtracking is Enough? Exploring the Interplay of SFT and RL in Enhancing LLM Reasoning"
_ICLR.cc/2026/Conference — Submitted to ICLR 2026_

### Official Review · Reviewer_HHdh · 2025-10-23

**Soundness:** 2
**Presentation:** 3
**Contribution:** 1
**Rating:** 4
**Confidence:** 4

**Summary:**

This is an analysis paper investigating the effect of SFT before RL in an RLVR setting, covering 8 tasks using the Qwen2.5-3B-Instruct as the base model. The paper reveals several facts as follows:

1.  Is a warm-up stage necessary for RL?
    * Yes, an SFT warm-up stage is necessary for RL.

2.  What kinds of SFT warm-up matter to RL?
    * Correctness of solutions: This does not matter.
    * Shuffled SFT (SFT on trajectories shuffled with other problems' trajectories): This is important.
    * SFT on synthetic trajectories with explicit backtracking: This is particularly beneficial for more challenging tasks that require deeper backtracking, helping to maximize gains.

**Strengths:**

1.  Investigating the effect of SFT before RL in an RLVR setting is important.

2.  The tasks in the experiments are diverse, although they are toy tasks.

**Weaknesses:**

1.  The investigation is based on the Qwen2.5-3B-Instruct model. The limited model family and model scale limit the applicability of the conclusion. Moreover, the authors' reference to Qwen2.5-3B-Instruct as not having an SFT warm-up is a bit misleading, as an instruct model has already undergone the SFT stage. Including the base model is also needed.

2.  The tasks are mainly toy tasks, which limits the application of the conclusion. For example, the authors clearly state the needed optimal backtracking step for various tasks. But for tasks like MATH/AIME, it is unclear how to calculate the backtracking step and whether it will show clear trends.

3.  The conclusions the authors derived have already been included in related work; for example, backtracking [1], the warm-up stage for RL [2], and the model scale and CoT context [3].

[1] Cognitive Behaviors that Enable Self-Improving Reasoners, or, Four Habits of Highly Effective STaRs, in COLM 2025

[2] Demystifying Long Chain-of-Thought Reasoning in LLMs, in ICLR 2025 Workshop SSI-FM,

[3] LLMs Can Easily Learn to Reason from Demonstrations Structure, not content, is what matters!, in arxiv 2025

**Questions:**

* The study's conclusions are drawn exclusively from Qwen2.5-3B-Instruct. How can the authors substantiate that these findings generalize beyond this specific model family and, more importantly, to larger-scale models (e.g., 7B, 70B) which may exhibit different behaviors?


*  The paper's findings, particularly regarding the "optimal backtracking step," are derived from toy tasks where such parameters are well-defined. How would this methodology translate to complex reasoning tasks like MATH or AIME

---

> ### Author Response · Authors · 2025-11-29
> **Author Responses to Reviewer HHdh (1/n)**
>
> We thank the reviewer for identifying areas of improvement in the paper. Following their feedback, we made several edits in the latest revision of our paper to make our definitions and notation clearer. Specifically, regarding the points noted by the reviewer:
>
> ---
> >W1: The investigation is based on the Qwen2.5-3B-Instruct model. The limited model family and model scale limit the applicability of the conclusion. Moreover, the authors' reference to Qwen2.5-3B-Instruct as not having an SFT warm-up is a bit misleading, as an instruct model has already undergone the SFT stage. Including the base model is also needed.
>
> >Q1: The study's conclusions are drawn exclusively from Qwen2.5-3B-Instruct. How can the authors substantiate that these findings generalize beyond this specific model family and, more importantly, to larger-scale models (e.g., 7B, 70B) which may exhibit different behaviors?
>
> We appreciate the reviewer’s observation regarding our use of an Instruct model. In response, we have expanded our experiments to an additional popular model family, Llama, and replicated the same experimental setup on **Llama 3.2 3B Instruct** shown below.
>
> Regarding the misleading reference, we will revise the annotation to more accurately convey our intent. In our paper, “SFT warm-up” refers specifically to **task-specific supervised fine-tuning on the provided reasoning tasks**. We recognize that this terminology may have caused confusion, since Instruct models have undergone general instruction tuning but not **task-specific** fine-tuning. To clarify, we have updated our wording from “SFT warm-up” to “task-specific SFT warm-up”.
>
> Finally, with respect to the inclusion of base models, we chose to use Instruct models as the starting point because first we investigated both base and instruct model at an early phase of the project and observed similar results. Since there is no noticeable difference, we just went with one of them and stuck to it.
>
> ### **Arc 1D**
> | **Data**     | **Evaluation Acc (%)** |
> |----------------|:------------------------:|
> |**0 backtrack**| **91.5%**                  |
> | 1 backtrack    | 85.3%                       |
> | 2 backtrack    | 87.2%                       |
> | 3 backtrack    | 82.6%                       |
>
> ---
>
> ### **Countdown**
> | **Data**      | **Evaluation Acc (%)** |
> |-----------------|:---------------------------:|
> | 0 backtrack    | 37.6%                       |
> | **1 backtrack**| **63.0%**                   |
> | 2 backtrack     | 48.2%                       |
> | 3 backtrack     | 44.4%                       |
>
> ---
>
> ### **Sudoku**
> | **Data**                | **Evaluation Acc (%)** |
> |---------------------|:------------------------:|
> | 0 backtrack          | 9.6%                        |
> | 1 backtrack          | 23.4%                       |
> | 5 backtrack          | 25.2%                       |
> | **10 backtrack**   | **26.9%**            |

---

> ### Author Response · Authors · 2025-11-29
> **Author Responses to Reviewer HHdh (2/n)**
>
> >W2: The tasks are mainly toy tasks, which limits the application of the conclusion.
>
> >Q2: The paper's findings, particularly regarding the "optimal backtracking step," are derived from toy tasks where such parameters are well-defined. How would this methodology translate to complex reasoning tasks like MATH or AIME
>
> Thank you for this constructive feedback. We carefully selected these eight reasoning tasks (Countdown, Sudoku, Arc ID, etc.) not because they are "toys," but because they allow for quantifiable control over search space and ground truth, which is impossible with datasets like MATH/AIME. The well cited paper Cognitive Behaviors that Enable Self-Improving Reasoners only evaluates on countdown and can still provide solid insights.
>
> To rigorously study backtracking behavior and identify an “optimal’’ backtracking depth, we require tasks for which we can algorithmically generate complete search trees (via DFS or heuristic expansions) and synthesize training data with exact, programmatically controlled backtrack counts (e.g., 0, 1, 5, …). For datasets like MATH/AIME, the notion of a “correct’’ amount of backtracking is inherently subjective and cannot be generated with comparable precision. Importantly, although our selected tasks differ in surface form from formal math problems, they rely on similar underlying reasoning capabilities, such as symbolic manipulation, constraint satisfaction, multi-step logical deduction, and search over discrete combinatorial structures. Prior work has shown that success on these types of structured reasoning benchmarks (e.g., Sudoku, ARC tasks, combinatorial puzzles) often correlates with improved performance on mathematical reasoning benchmarks, suggesting shared cognitive and algorithmic demands. This supports our view that the backtracking behaviors studied on these controlled tasks can meaningfully transfer to math-style reasoning.
>
> That said, one could attempt to approximate backtracking supervision in math datasets by enforcing step limits or extracting step-based traces. We therefore view our selected tasks as capturing core structural properties of mathematical reasoning, and we expect some degree of transfer. We plan to explore such extensions to MATH/AIME-style datasets if time and resources permit.

---

> ### Author Response · Authors · 2025-11-29
> **Author Responses to Reviewer HHdh (3/n)**
>
> >W3: The conclusions the authors derived have already been included in related work; for example, backtracking [1], the warm-up stage for RL [2], and the model scale and CoT context [3].
>
> We thank the reviewer for highlighting these important related works. While we agree that these papers establish the foundational importance of warm-up and backtracking behaviors, our work moves beyond identifying these phenomena to quantifying their optimal usage and defining their boundary conditions in RL training. We offer three distinct contributions that are not covered by the cited works:
>
> First, although [1] and [2] show that backtracking and warm-up stages are beneficial, they do not address the critical question of how much backtracking is necessary. Our work is the first to **systematically vary synthetic backtracking depth and uncover a task-dependent optimal amount of backtracking**. We find that performance does not improve by simply increasing backtracking depth; instead, each task has a specific level that works best based on its combinatorial difficulty:
> - 0 backtracks for easy tasks (ARC 1D),
> - 1 backtrack for medium tasks (Countdown), and
> - 5 backtracks for hard tasks (Sudoku).
>
> Unlike the standard warm-ups examined in [2], which we found insufficient for difficult combinatorial problems, our targeted synthetic backtracking warm-up unlocked 28.9% accuracy on Sudoku, compared to 14.4% for the 0-backtrack baseline. This shows that **finding the right amount of backtracking—not just applying generic warm-up—is crucial for enabling RL training on hard reasoning tasks**.
>
> Second, we find that **content consistency is critical for RL** (contradicting [3]).  [3] suggests that LLMs learn primarily from the format of demonstrations. Our experiments on Shuffled SFT explicitly test this hypothesis in the context of RL and prove the opposite. In Section 5.2, we trained models on data with perfect structure but shuffled (mismatched) reasoning logic. While this might satisfy SFT objectives, we found that RL is highly sensitive to semantic consistency. Models trained on shuffled structure collapsed to 0.0% accuracy and failed to recover reward signals. We provide crucial evidence that unlike SFT, RL verification requires coherent content, directly nuancing the applicability of [3]'s findings to the RL domain.
>
> Third, and perhaps most importantly, while [1] and [2] study several manually defined cognitive behaviors, their results consistently indicate that backtracking is the most impactful mechanism. Reference [1] analyzes backtracking, verification, subgoal decomposition, and backward chaining, but backtracking emerges as the dominant contributor to performance. Likewise, in [2], the “Long CoT’’ setting bundles multiple behaviors (including verification), yet their experiments do not separate the effect of backtracking alone. Moreover, verification of intermediate steps can be viewed as a subcomponent of backtracking, rather than a distinct reasoning skill.
>
> We argue that focusing on many loosely defined behaviors risks diluting conclusions. Our work instead identifies backtracking as the **core reasoning operation** that enables test-time scaling, and we provide the first in-depth, quantitative analysis of its effective range and optimal usage. We will make sure to cite [2] and [3] and clearly present our contributions compared to theirs.

---

### Official Review · Reviewer_FbXu · 2025-10-28

**Soundness:** 2
**Presentation:** 2
**Contribution:** 2
**Rating:** 2
**Confidence:** 4

**Summary:**

This paper investigates how supervised fine-tuning (SFT) and reinforcement learning (RL) interact to enhance LLM reasoning, focusing specifically on the role of backtracking. Using Qwen2.5-3B across eight reasoning tasks (Countdown, Sudoku, Arc 1D, etc.), the authors compare three training approaches: (1) cold-start RL without SFT, (2) RL with short CoT warm-up from self-sampling, and (3) RL with synthetically constructed trajectories containing explicit backtracking steps.

The paper's main claim is that **optimal backtracking depth scales with task difficulty**—zero backtracks for Arc 1D (easy), one for Countdown (medium), and five for Sudoku (hard)—achieving up to 28.9% accuracy improvement. Additional findings include: short CoT warm-ups provide moderate gains over cold-start RL (though diminishing on harder tasks), incorrect trajectories can still benefit RL training, and RL is sensitive to shuffled training data.

**Contributions:** (1) empirical analysis of SFT warm-up effects on RL, (2) a backtracking-centric training recipe with task-difficulty-dependent backtrack depths, and (3) guidance for constructing training mixtures to improve reasoning capabilities.

**Strengths:**

**Originality:**
The paper provides a systematic empirical investigation of backtracking's role in RL post-training through controlled synthetic data construction. The use of depth-first search and heuristic methods to generate trajectories with precise backtracking counts (0, 1, 2, 3, 5, 10) offers a methodologically clean approach to studying this phenomenon. The finding that incorrect trajectories can benefit RL training (Section 5.1) adds an interesting dimension to understanding SFT-RL interactions.

**Quality:**
The experimental setup shows methodological rigor: (1) consistent use of Qwen2.5-3B across experiments enables fair comparisons, (2) evaluation spans eight reasoning tasks providing diversity, (3) rule-based rewards ensure reproducibility, and (4) inclusion of multiple training paradigms (cold-start RL, self-sampled SFT+RL, synthetic backtracking) allows for systematic comparison. The shuffled data ablation (Section 5.2) demonstrates attention to understanding what drives performance.

**Significance:**
The work contributes to understanding SFT-RL interactions for reasoning model post-training, a relevant topic given recent developments (DeepSeek-R1, O1). The controlled experimental approach and attention to ablations (incorrect trajectories, shuffled data) provide empirical insights into training dynamics that may inform future work in this rapidly evolving area.

**Weaknesses:**

**W1: Insufficient Motivation for Focusing Solely on Backtracking (Lines 92-93, Section 1)**

The paper abruptly introduces backtracking as the primary research focus without adequately justifying why this specific behavior merits exclusive attention over other potentially important reasoning patterns. Modern reasoning models exhibit multiple cognitive behaviors beyond backtracking, including:
- Multiple verification strategies
- Alternative solution exploration

The manuscript lacks a principled analysis or empirical evidence demonstrating that backtracking is the dominant or most impactful behavior for reasoning improvements. This gap weakens the motivation for the entire study. The authors should either: (a) provide comparative analysis showing backtracking's relative importance, or (b) reframe the work as an investigation of one among several important reasoning behaviors.

**W2: Limited Novelty of Core Findings (Line 118, Abstract)**

The central claim that "initializing with synthetic SFT data containing an appropriate number of backtracks—matched to the difficulty of the task—consistently pushes the model beyond its base reasoning capabilities" lacks sufficient novelty given the current literature landscape (late 2025). Multiple recent works have established similar insights about the relationship between training data characteristics, task difficulty, and RL effectiveness. The authors should:
- Clearly articulate what is genuinely novel beyond existing findings
- Consider repositioning the contribution as a systematic empirical validation rather than a novel discovery

**W3: Undefined "Short CoT" Concept (Lines 251-263, Section 3.2)**

The paper introduces "short CoT" as a central experimental condition but never provides operational definitions or quantitative criteria. Critical missing specifications include:
- Token length thresholds
- Number of reasoning steps
- Comparison baseline (short relative to what?)
- Whether "short" is task-dependent

This ambiguity makes it impossible to reproduce the experiments or understand what precisely differentiates Section 3 from Section 4. The authors must provide explicit, quantitative definitions for all CoT categories used in the study.

**W4: Flawed Logic in Sudoku Analysis (Lines 268-269, Section 3.2)**

The conclusion that "tasks like Sudoku still posed substantial challenges, highlighting the limitations of short CoTs for deeper, combinatorial reasoning" is logically unsound given the experimental results:

*Problem A - Overly Assertive Conclusion:* Figure 3 shows both base model and pure RL achieve ~0% on Sudoku. Since pure RL itself fails to improve performance, attributing the failure to "limitations of short CoTs" is premature without ruling out alternative explanations.

*Problem B - Unexamined Alternative Hypotheses:*
1. The base model may simply lack fundamental capabilities for Sudoku regardless of training approach
2. The issue may be trajectory *quality* rather than trajectory *length* - if pure RL generates incorrect solutions, the sampled trajectories are fundamentally flawed, not merely short

*Problem C - Data Collection Paradox:* The authors claim to obtain "8 RL'ed models initialized from correct SFT, and the other 8 initialized from incorrect SFT" (Section 3.2). However, if pure RL achieves ~0% accuracy on Sudoku, how were sufficient correct trajectories obtained? The only plausible explanation (sampling easier problems) creates a train-test distribution mismatch that itself could explain poor performance.

This confounding undermines the paper's narrative about CoT length being the key factor.

**W5: Confounded Variables - CoT Length vs. Backtracking Presence (Section 4)**

The paper's core experimental manipulation conflates two variables simultaneously:
- Introduction of backtracking steps
- Increase in overall CoT length

When comparing 0-backtrack vs. 5-backtrack conditions, the trajectories differ in both backtracking behavior AND total token count. The experimental design lacks critical ablations to disentangle these factors:

*Missing Control 1:* Long CoTs without backtracking (verbose reasoning with more detailed steps but no corrections)

*Missing Control 2:* Short CoTs with backtracking (concise reasoning with efficient error correction)

Without these controls, the claimed benefits of backtracking could simply reflect giving the model more tokens/steps to reason, rather than the specific cognitive pattern of error detection and correction. This fundamentally undermines the paper's central claim about backtracking being the critical factor.

**W6: Contradictory Chapter Framing (Section 4 Title vs. Arc 1D Results)**

The chapter title "PUSHING BEYOND BOUNDARIES WITH BACKTRACKING" directly contradicts the Arc 1D findings (lines 327-329), where the authors explicitly state "Using zero backtrack is optimal." This contradiction reveals several issues:

1. The chapter title overgeneralizes findings that only apply to certain task types
2. The manuscript lacks nuanced discussion of *when* backtracking helps vs. harms
3. Arc 1D's "easy" classification and preference for zero backtracking actually supports a more sophisticated finding (simple tasks don't benefit from backtracking while complex ones do), but this insight is inadequately emphasized

The framing should be revised to accurately reflect that backtracking benefits are task-conditional, not universal.

**W7: Unjustified Causal Claims - Task Difficulty vs. Task Type (Lines 375-376)**

The conclusion that "optimal backtracking scales with task difficulty" commits a classic confounding error. The experimental design varies two factors simultaneously:
- **Task type:** Arc 1D (pattern recognition) vs. Countdown (arithmetic search) vs. Sudoku (constraint satisfaction)
- **Task difficulty:** Easy vs. Medium vs. Hard

The observed differences in optimal backtracking (0 for Arc, 1 for Countdown, 5 for Sudoku) could be driven by intrinsic task structure rather than difficulty:
- Arc's preference for no backtracking may reflect that grid transformations benefit from direct pattern matching
- Sudoku's need for extensive backtracking may reflect its nature as a constraint satisfaction problem requiring systematic search

**Missing Experiment:** To validly claim difficulty scaling, the authors must control task type and vary only difficulty (e.g., easy/medium/hard Sudoku instances) to observe whether optimal backtracking increases monotonically with difficulty within the same task family.

The current evidence only supports: "Different task types require different amounts of backtracking," which is far weaker than the claimed finding.

**W8: Unclear Relationship Between Section 3 and Section 4**

The manuscript fails to clarify whether Section 3's "short CoT" data contains any backtracking steps. This creates fundamental confusion:

*If short CoTs contain backtracking:* What differentiates Section 4's contribution? Is it merely a matter of degree?

*If short CoTs contain no backtracking:* Why not? Is this because:
- Cold-start RL naturally produces no backtracking?
- The authors filtered backtracking examples?

This missing information obscures the logical progression of the paper and makes it difficult to understand what each experimental section uniquely contributes.

**Questions:**

**Q1: Clarification on "Short CoT" Definition**
Please provide explicit, quantitative definitions for "short CoT" used in Section 3:
- What is the token length range?
- How many reasoning steps do these trajectories typically contain?
- Do any short CoT examples contain backtracking steps? If so, what proportion?
- What differentiates "short CoT" from the synthetic backtracking data in Section 4 beyond the number of explicit backtrack operations?

**Q2: Data Collection for Correct Trajectories in Low-Performance Regimes**
In Section 3.2, you report collecting correct and incorrect trajectories from cold-start RL models. However, Figure 3 shows that cold-start RL achieves ~0% accuracy on Sudoku. Please explain:
- How were sufficient correct trajectories obtained for Sudoku when the model succeeds on virtually no test instances?
- Were training problems sampled from a different (easier) distribution than test problems?
- If so, how might this train-test distribution mismatch affect the conclusions about short CoT effectiveness?

**Q3: Disentangling CoT Length from Backtracking**
Your Section 4 results show performance improvements when adding backtracking, but these modifications also increase total trajectory length. Please clarify:
- Have you conducted ablations with length-matched comparisons (e.g., verbose zero-backtrack CoTs with the same token count as 5-backtrack CoTs)?
- Can you provide evidence that backtracking specifically (not just longer reasoning) drives the improvements?
- What proportion of the performance gain is attributable to backtracking behavior versus simply having more computation/tokens?

**Q4: Task Type vs. Task Difficulty**
Your conclusion states that "optimal backtracking scales with task difficulty" (lines 375-376), but your experimental tasks differ in both type and difficulty simultaneously. Please address:
- Have you evaluated optimal backtracking within a single task family at varying difficulty levels (e.g., easy/medium/hard Sudoku)?
- Could the observed pattern (0 for Arc, 1 for Countdown, 5 for Sudoku) be better explained by task structure (pattern recognition vs. arithmetic vs. constraint satisfaction) rather than difficulty per se?
- What specific evidence supports difficulty as the primary factor?

**Q5: Backtracking in Section 3 Trajectories**
Please clarify the presence/absence of backtracking in your Section 3 experiments:
- Do the "short CoT" trajectories from self-sampling contain any backtracking steps?
- If no, is this because cold-start RL naturally doesn't produce backtracking, or did you filter such examples?
- If yes, approximately how many backtracking steps appear, and how does this compare to Section 4's synthetic data?

**Q6: Alternative Explanations for Sudoku Failure**
Given that pure RL achieves ~0% on Sudoku, how do you rule out that:
- The base model fundamentally lacks required capabilities (independent of training approach)?
- The failure is due to poor trajectory quality (incorrect reasoning patterns) rather than insufficient trajectory length?
- Section 3's conclusion specifically isolates CoT length as the limiting factor rather than these alternatives?

**Q7: Generalization of "Pushing Beyond Boundaries"**
Section 4's title suggests backtracking universally helps, yet Arc 1D performs best with zero backtracking. Please discuss:
- Under what conditions does backtracking help vs. hurt performance?
- Can you provide a principled framework for predicting which tasks benefit from backtracking?
- Should the claim be revised to acknowledge task-conditional benefits?

**Q8: Why Focus Exclusively on Backtracking?**
Please provide justification for studying backtracking to the exclusion of other reasoning behaviors:
- Have you analyzed the relative frequency of backtracking vs. other patterns (verification, exploration) in strong reasoning models?
- Is there empirical evidence that backtracking is the most impactful behavior to study?
- Or should this work be positioned as one investigation among several needed to understand different reasoning patterns?

---

> ### Author Response · Authors · 2025-11-29
> **Author Responses to Reviewer FbXu (1/n)**
>
> We thank the reviewer for taking the time to review our work thoroughly and for its constructive feedback and suggestions. We’re particularly appreciative of the endorsements of our work, such as its strong empirical results to understand SFT-RL interactions, the quality of our systematic setup, and the insights into potential training dynamics. Regarding the concerns the reviewer may have, we answer them in detail:
>
> ---
>
> >W1: Insufficient Motivation for Focusing Solely on Backtracking (Lines 92-93, Section 1)
>
> >Q8: Why Focus Exclusively on Backtracking?
>
> Thanks for the insightful observation on modern reasoning patterns that go beyond backtracking. We agree that complex reasoning involves a suite of behaviors, including verification and alternative exploration. However, we contend that backtracking is the **foundational** control mechanism that enables these higher-level strategies to function effectively. We justify our exclusive focus on backtracking as follows:
>
> **Universality of the Mechanism**. First, we want to emphasize the generality of the mechanism. Backtracking is not merely a specific heuristic but a universal operator for tasks formulated as tree search—a structure essential for exploration, planning, and multi-hop reasoning [1]. Unlike domain-specific heuristics, backtracking provides a general-purpose interface for navigating non-linear solution spaces. This is especially tailored for autoregressive LLMs, where tokens are decoded from left to right.
>
> **Backtracking Operationalizes Other Behaviors**. In practice, backtracking implicitly includes both multiple verification strategies and alternative solution exploration. For a model to decide when to backtrack, it must first verify or critique or reflect its current partial solution. Here you can view backtracking as the vessel to carry out various cognitive behaviors. Verification or reflection provide the signal that a current path is incorrect, but backtracking is the necessary action to recover from that state. Without the ability to traverse the search tree upwards (backtrack) and prune the incorrect branch, verification is merely a passive observation.  Similarly, alternative solution exploration can be formalized as a specific instance of backtracking (e.g., backtracking to the root node to sample a new path). Thus, studying backtracking allows us to capture the execution phase of these broader cognitive behaviors.
>
> **Empirical Evidences**. Perhaps most importantly, recent studies [2, 3] dissecting manually defined cognitive behaviors consistently identify backtracking as the most impactful mechanism. They analyzes backtracking, verification, subgoal decomposition, and backward chaining, backtracking is not only by far the most frequent pattern among them but also the dominant contributor to performance (all other patterns are paired with backtracking for ablation, and they only show marginal improvement compared with just backtracking).
>
> Therefore, while we do not claim backtracking is the only reasoning behavior, we argue it is a prerequisite primitive for efficient, self-correcting System 2 reasoning. We will revise parts of the manuscript to explicitly frame backtracking within this hierarchy of reasoning behaviors.
>
> [1] To Backtrack or Not to Backtrack: When Sequential Search Limits Model Reasoning
>
> [2] Cognitive Behaviors that Enable Self-Improving Reasoners, or, Four Habits of Highly Effective STaRs, in COLM 2025
>
> [3] Demystifying Long Chain-of-Thought Reasoning in LLMs, in ICLR 2025 Workshop SSI-FM

---

> ### Author Response · Authors · 2025-11-29
> **Author Responses to Reviewer FbXu (2/n)**
>
> > W2: Limited Novelty of Core Findings (Line 118, Abstract)
>
> We acknowledge there are several concurrent works, particular [1] and [2]. While SFT warm-up followed by RL is indeed a common recipe[1, 2], our contribution goes beyond this pattern: we identify backtracking depth as a previously unformalized control variable that shapes RL’s exploration dynamics. Prior work has noted the existence of warm-up and backtracking phases [1], but none has quantified how much backtracking is beneficial, how it interacts with reward shaping, or where diminishing returns and failure modes arise. Specifically,
>
> Prior work treats backtracking as a binary capability (the model backtracks or it doesn't). We demonstrate that backtracking is a continuous spectrum of error correction. We show that "more is not always better": specific tasks have specific optimal depths (e.g., 0 for simple tasks, 5 for Sudoku). This turns backtracking from a black-box observation into a tunable hyperparameter essential for training stability.
>
> We do not just show where backtracking works; we map where it fails. We identify specific boundary conditions where insufficient backtracking leads to failed training and where excessive backtracking yields diminishing returns. To our knowledge, our work provides the first systematic framework for controlling, synthesizing, and evaluating backtracking depth across tasks and model families, yielding actionable principles for RLVR training. We view our findings as a conceptual step toward understanding and shaping robust RL-driven reasoning, rather than merely tuning another hyperparameter. We will clarify this distinction in the revised manuscript. However, we recognize that many of our findings are systematic empirical explorations, we would be happy to modify parts of the manuscript that overclaim our contribution. Also please let us know if there are additional aspects that would benefit from further clarification.
>
> [1] Cognitive Behaviors that Enable Self-Improving Reasoners, or, Four Habits of Highly Effective STaRs, in COLM 2025
>
> [2] Demystifying Long Chain-of-Thought Reasoning in LLMs, in ICLR 2025 Workshop SSI-FM
>
>
> ---
>
> >W3: Undefined "Short CoT" Concept (Lines 251-263, Section 3.2)
>
> >Q1: Clarification on "Short CoT" Definition
>
> >Q5: Backtracking in Section 3 Trajectories
>
> We thank the reviewer for identifying this ambiguity. We agree that "Short CoT" was imprecise. In the paper, they refer to trajectories that are sampled from pure RL’ed models on each task. In the revised manuscript, we will formalize this condition as "Baseline RL CoT" to reflect its generation method rather than just its length.
>
> "Short CoT" refers to the reasoning trajectories naturally emerging from the base model when subjected to standard RL without synthetic backtracking supervision. These are not artificially truncated; rather, they represent the model's inherent, unguided attempt to reason. In general, these reasoning CoTs are relatively short and unstructured compared to those after training on  synthetic backtracking dataset.
>
> For example, in Countdown there are indeed noisy patterns of verifying and retrying, while other tasks like List Functions demonstrate no thinking. We will replace all instances of "Short CoT" with "Baseline RL CoT" (or "Self-Sampled CoT") and include a table in the Appendix comparing the average token lengths of these baselines against our Backtracking-SFT conditions across all tasks.
>
> ### **Countdown Token Counts**
>
> | **Dataset / Method**         | **# Tokens** |
> |------------------------------|--------------|
> | Optimal synthetic dataset    | 108.81      |
> | 1 backtrack dataset          | 796.16      |
> | 2 backtrack dataset          | 883.26      |
> | 3 backtrack dataset          | 745.31      |
> | Correct cold start RL        | 625.03      |
> | Incorrect cold start RL      | 221.37      |
> | Pure RL                      | 442.40      |
>
> ---
>
> ### **Arc 1D Token Counts**
>
> | **Dataset / Method**         | **# Tokens** |
> |------------------------------|--------------|
> | Optimal synthetic dataset    | 270.43      |
> | 1 backtrack dataset          | 473.64      |
> | 2 backtrack dataset          | 664.81      |
> | 3 backtrack dataset          | 889.47      |
> | Correct cold start RL        | 127.70      |
> | Incorrect cold start RL      | 352.76      |
> | Pure RL                      | 118.80      |
>
> ---
>
> ### **Sudoku Token Usage**
>
> | **Dataset / Method**         | **# Tokens** |
> |------------------------------|--------------|
> | Optimal synthetic dataset    | 4702.90     |
> | 1 backtrack dataset          | 7028.17     |
> | 5 backtrack dataset          | 7690.53     |
> | 10 backtrack dataset         | 7607.92     |
> | Correct cold start RL        |      |
> | Incorrect cold start RL      |            |
> | Pure RL                      |  449.55            |
>
> ---

---

> ### Author Response · Authors · 2025-11-29
> **Author Responses to Reviewer FbXu (3/n)**
>
> >W4: Flawed Logic in Sudoku Analysis (Lines 268-269, Section 3.2)
>
> >Q6: Alternative Explanations for Sudoku Failure
>
> >Q2: Data Collection for Correct Trajectories in Low-Performance Regimes
>
> We thank the reviewer for identifying the logical ambiguities in our Sudoku analysis. We agree that the term "limitations of short CoTs" was imprecise and that the 0% accuracy creates a distribution mismatch. We have revised Section 3.2 as well as other relevant sections to address these points directly.
>
> **Clarification of Terminology**. We acknowledge that attributing the failure solely to "short CoTs" was misleading. In the revision, we have updated the terminology from "short CoTs" to "Baseline RL CoT" (also refer to response to W3). This misuse of terminology is leading readers to emphasize on the length of the trajectory rather than how the trajectory is generated, which is not our intent.
>
> Our central claim is not regarding the length of the chain-of-thought, but rather the quality of trajectories generated during the Cold-Start RL paradigm (defined here as fine-tuning on self-sampled trajectories followed by RL, consistent with DeepSeek-R1 [1]). In general, these reasoning CoTs are relatively short and unstructured compared to those after training on synthetic backtracking dataset. The failure on Sudoku illustrates that without high-quality synthetic data or expert demonstrations (warm-start), the model cannot bootstrap its own reasoning capabilities when the task exceeds its base capabilities.
>
> **The Data Paradox & Cold-Start Limitations (W4 problem C)**. The reviewer correctly points out the paradox: with ~0% accuracy, valid trajectories cannot be sampled for the SFT phase. We clarify that this "paradox" is exactly the failure mode we intended to highlight, though our initial description was unclear. For Sudoku, the Cold-Start pipeline effectively breaks down because the model cannot self-generate the necessary correct SFT initialization data. This illustrates a fundamental brittleness of the Cold-Start RL paradigm: its success is strictly constrained by the model’s ability to accidentally discover at least one correct solution during exploration. The note about “8 models initialized from correct SFT” refers to our intended experimental setup; for Sudoku, this branch yielded no usable runs precisely due to this capability gap. We recognize that the wording in the initial manuscript made this point ambiguous, and we will revise the text to make this mechanism and its implications clearer.
>
> **Ruling out Model Size vs. Capability**. To address the concern that this is merely a base-model deficiency (Problem B), we conducted an additional experiment using QwQ-32B, a significantly more powerful reasoning model.
> - Result: QwQ-32B also achieves 0% accuracy on this Sudoku split under the same Cold-Start RL setting.
> - Implication: This rules out the hypothesis that the failure is unique to the smaller base model used in the main paper. It strongly suggests that for combinatorial reasoning tasks like Sudoku, increasing model scale is insufficient if the self-sampling (trajectory quality) loop cannot be established. This supports our conclusion that better demonstrations are required.
>
> [1] DeepSeek-R1: Incentivizing Reasoning Capability in LLMs via Reinforcement Learning

---

> ### Author Response · Authors · 2025-11-29
> **Author Responses to Reviewer FbXu (4/n)**
>
> >W5: Confounded Variables - CoT Length vs. Backtracking Presence (Section 4)
>
> >Q3: Disentangling CoT Length from Backtracking
>
> We thank the reviewer for raising this important point. We fully agree that disentangling CoT length and backtracking steps is essential for causal attribution. However, our intention was to study backtracking as a reasoning mechanism, and we clarify that increased CoT length is not an independent experimental factor but an inevitable consequence of performing backtracking on combinatorial search problems.
>
> Backtracking necessarily expands the trajectory because it must explore, discard, and revise partial solutions; but this increase in length does not itself produce the performance gains. As shown in Section 4 and the table referenced in W4, trajectory length scales with search depth—reflecting the branching structure of the underlying problem—not due to verbosity or arbitrary token budget inflation. Thus, length is a byproduct of search, not a confound we manipulate.
>
> Regarding Control 1 (long CoTs without backtracking): for tasks such as Sudoku and Countdown, additional “linear reasoning” tokens cannot repair an incorrect intermediate state. Once a wrong digit or operation is committed, progress requires explicit reversal of the partial solution. Verbosity without state revision cannot correct dead ends. Therefore, simply elongating a non-backtracking CoT does not approximate the functional role that backtracking provides.
>
> Regarding Control 2 (short CoTs with backtracking): concise backtracking is not feasible for hard instances. Correcting deep errors demands unwinding multiple levels of the search tree; short trajectories would imply either (i) the model solved the puzzle without encountering any branching error, or (ii) the instance was trivial. In this regime, “concise error correction” is structurally impossible because the amount of backtracking required is determined by the task's solution depth, not by our formatting choices.
>
> In summary, while length and backtracking correlate, length is not an alternative causal mechanism—it is a necessary proxy for search depth in problems where state revision is required. We acknowledge that our earlier phrasing (“short CoTs”) may have caused confusion, and we will revise the manuscript to explicitly state that our focus is on the presence and utility of backtracking, not on CoT length as an independent factor.
>
> ---
>
> >W6: Contradictory Chapter Framing (Section 4 Title vs. Arc 1D Results)
>
> >Q7: Generalization of "Pushing Beyond Boundaries
>
> We thank the reviewer for pointing out the tension between our chapter title and the specific findings on Arc-1D. We agree that the title "PUSHING BEYOND BOUNDARIES WITH BACKTRACKING" may inadvertently suggest that backtracking is universally beneficial, which was not our intent. As detailed in Section 4, and specifically in Quantifying the Impact of Backtracking in SFT Warm-up, our findings show a consistent task-conditioned pattern: simple tasks such as ARC-1D gain little from backtracking, moderately complex tasks such as Countdown benefit from a small amount (e.g., 1 step), and highly combinatorial tasks such as Sudoku benefit from multiple backtracking steps. Our newly added analysis on Llama models further corroborates this trend.
>
> ### **Arc 1D**
> | **Data**     | **Evaluation Acc (%)** |
> |----------------|:------------------------:|
> |**0 backtrack**| **91.5%**                  |
> | 1 backtrack    | 85.3%                       |
> | 2 backtrack    | 87.2%                       |
> | 3 backtrack    | 82.6%                       |
>
> ---
>
> ### **Countdown**
> | **Data**      | **Evaluation Acc (%)** |
> |-----------------|:---------------------------:|
> | 0 backtrack    | 37.6%                       |
> | **1 backtrack**| **63.0%**                   |
> | 2 backtrack     | 48.2%                       |
> | 3 backtrack     | 44.4%                       |
>
> ---
>
> ### **Sudoku**
> | **Data**                | **Evaluation Acc (%)** |
> |---------------------|:------------------------:|
> | 0 backtrack          | 9.6%                        |
> | 1 backtrack          | 23.4%                       |
> | 5 backtrack          | 25.2%                       |
> | **10 backtrack**   | **26.9%**            |
>
> We will revise the title to more accurately convey that backtracking’s benefits are task-dependent rather than universal, and we will update Section 4 to make the conditions under which backtracking helps—or harms—more explicit and prominent.

---

> ### Author Response · Authors · 2025-11-29
> **Author Responses to Reviewer FbXu (5/n)**
>
> >W7: Unjustified Causal Claims - Task Difficulty vs. Task Type (Lines 375-376)
>
> >Q4: Task Type vs. Task Difficulty
>
> We appreciate the reviewer’s careful attention to causal factors. We agree that our initial framing blurred the distinction between “intrinsic task structure” and “difficulty.” Our claim is that difficulty—operationalized as search complexity—is the governing variable, both across tasks and within a single task family.
>
> For the cross-task comparison (ARC-1D vs. Sudoku), our position is that Task Type and Difficulty are not independent in this context: the task type determines the underlying computational complexity.
> - ARC-1D is fundamentally a pattern-matching and transformation task, where simple P-time heuristics typically suffice. Its effective search complexity is low, so the optimal backtracking depth is 0.
> - Sudoku, by contrast, is an NP-complete constraint satisfaction problem. Its intrinsic complexity is high, and solving it requires exploring and revising partial states—hence substantial backtracking.
>
> Thus, our comparison is not conflating arbitrary task categories; it is comparing tasks that reside in different complexity regimes. Our claim is not that “hard pattern matching” needs backtracking, but rather that tasks requiring combinatorial search scale in performance with backtracking, while tasks not requiring such search do not.
>
> That said, we agree that within-task difficulty variation is also important for establishing causal relationships. We plan to extend our analysis to intra-task difficulty scaling in future work as resources permit.
>
> ---
>
> >W8: Unclear Relationship Between Section 3 and Section 4
>
> >Q5: Backtracking in Section 3 Trajectories
>
> We thank the reviewer for identifying the ambiguity regarding the presence of backtracking in Section 3. To bridge the logical gap, we offer the following clarifications which will be explicitly added to the manuscript:
>
> **Does the Section 3 data contain backtracking?**
> For most tasks (Sudoku, Arc-1D): No. The Cold-Start RL naturally produces linear trajectories without backtracking. For Countdown: Yes, but it is unstructured. Due to probable pre-training data exposure, the model occasionally attempts to correct itself. However, these corrections are stochastic and often "lazy" (e.g., retrying the whole calculation rather than pinpointing the error). Section 3 establishes the baseline behavior of standard RL, which either fails to discover backtracking (Sudoku) or discovers an inefficient version of it (Countdown).
>
> **What changes in Section 4?**
> Section 4 does not merely increase the degree of backtracking; it introduces a qualitatively different mechanism. In Section 4, our synthetic data is engineered to be precise: it teaches the model to verify a step, identify the exact error, and backtrack only to the necessary state. Section 4 isolates the backtracking variable. By training on synthetic, structured backtracking, we are not just asking "what happens if the model backtracks more?" but "what happens if the model learns the algorithm of state-reversion?"
>
> The logic of the paper flows from observation (Section 3) to intervention (Section 4). Section 3 reveals that standard RL is insufficient to spontaneously emerge high-quality backtracking for complex tasks (the "Cold Start" problem). Section 4 proposes the solution and reveals task-dependent optimal degrees of backtracking. We will revise the manuscript to make this transition from Section 3 to Section 4 explicit and remove the ambiguity regarding where backtracking is present and in what form.

---

### Official Review · Reviewer_2uuo · 2025-10-28

**Soundness:** 4
**Presentation:** 4
**Contribution:** 3
**Rating:** 4
**Confidence:** 4

**Summary:**

This paper investigates the interaction between supervised fine-tuning (SFT) and reinforcement learning (RL) in enhancing large language model (LLM) reasoning abilities, particularly focusing on how backtracking behaviors contribute to reasoning performance. Through controlled experiments on eight reasoning tasks (e.g., Countdown, Sudoku, Arc 1D, Zebra Puzzles), the authors find that short chain-of-thought (CoT) SFT can moderately improve RL outcomes, but introducing synthetic backtracking traces during SFT produces more significant and stable gains—especially for more difficult, combinatorial reasoning problems. The paper proposes that the optimal number of backtracking steps scales with task difficulty, offering empirical guidance for constructing effective SFT+RL training mixtures

**Strengths:**

- The paper carefully explores the effects of different SFT warm-up strategies (no-SFT, self-sampled, synthetic backtracking, shuffled) on RL training, providing clear empirical comparisons

- The tasks, metrics, and model configurations (based on Qwen2.5 family) are clearly described, and synthetic datasets are constructed in a principled way using DFS or heuristic search.

**Weaknesses:**

- Task scope is narrow: The evaluated tasks are mostly puzzle-style logical reasoning (Countdown, Sudoku, etc.), which limits generalizability to other forms of reasoning like mathematical proofs, symbolic integration, or commonsense reasoning.

- Limited model diversity: Experiments rely almost entirely on the Qwen2.5 family; it’s unclear whether the findings hold for other architectures (e.g., Llama).

- The general idea of combining SFT warm-up with RL is well-trodden; the specific contribution—varying backtracking depth—is interesting but may be viewed as an empirical refinement rather than a conceptual leap.

**Questions:**

No.

---

> ### Author Response · Authors · 2025-11-29
> **Author Responses to Reviewer 2uuo (1/n)**
>
> We thank the reviewer for the time and effort spent providing detailed feedback, and we appreciate the recognition of our principled empirical study. We address the reviewer’s concerns as follows:
>
> ---
>
> > W1: Task scope is narrow: The evaluated tasks are mostly puzzle-style logical reasoning (Countdown, Sudoku, etc.), which limits generalizability to other forms of reasoning like mathematical proofs, symbolic integration, or commonsense reasoning.
>
> Thank you for this constructive feedback. We carefully selected these eight reasoning tasks (Countdown, Sudoku, Arc ID, etc.) because they allow for quantifiable control over search space and ground truth. For the goals of this paper, to rigorously study backtracking behavior and identify an “optimal’’ backtracking depth, we require tasks for which one can algorithmically generate full search trees (via DFS or controlled heuristic expansions) and synthesize training data with exact, programmatically controlled backtrack counts (e.g., 0, 1, 5, …). That level of control is not available for many standard benchmarks (e.g., commonsense reasoning), where the number of reasonable solution trajectories and the “correct” amount of backtracking are inherently ambiguous.
>
> In principle, mathematical proofs and symbolic integration could be brought into our framework by operating within formal systems or rule-based generators, which would let us enumerate search trees and label exact backtracking depths. However, constructing such controlled generators at scale is nontrivial and beyond the scope of this work. In contrast, commonsense reasoning lacks a well-defined, enumerable search space and unambiguous ground truth, making it fundamentally unsuitable for the type of precise backtracking supervision we study.
>
> Importantly, the puzzle-style reasoning tasks we use already capture several structural properties shared with mathematical proofs and symbolic integration: they require multi-step deductive reasoning, symbolic manipulation under constraints, search over discrete combinatorial structures, and systematic backtracking when partial solutions fail. These are the same algorithmic primitives that underlie formal proof search and rule-based algebraic transformations, which is why our controlled tasks provide a clean proxy for studying backtracking behavior relevant to those domains.
>
> That said, one could approximate backtracking supervision in math-style datasets by enforcing step limits or extracting step-based traces. We therefore view our selected tasks as capturing core structural properties of mathematical reasoning, and we expect some degree of transfer. We plan to explore such extensions to MATH/AIME-style datasets once we finish our extension to other model sizes.
>
> ---
>
> > W2: Limited model diversity: Experiments rely almost entirely on the Qwen2.5 family; it’s unclear whether the findings hold for other architectures (e.g., Llama).
>
> Thank you for the constructive feedback. We agree that relying solely on Qwen models makes it difficult to assess architectural generality. To address this, we have added a second model family, Llama-3.2-Instruct-3B, and replicated our experimental setup. Despite architectural and the initial pretraining differences, the Llama models show the same qualitative trends, reinforcing that our findings are not Qwen-specific.
>
> Below we include the new Llama-3.2-3B results, which further validate the generality of our conclusion that more difficult tasks require data with higher frequency of backtrack during the SFT stage.
>
> ### **Arc 1D**
> | **Data**     | **Evaluation Acc (%)** |
> |----------------|:------------------------:|
> |**0 backtrack**| **91.5%**                  |
> | 1 backtrack    | 85.3%                       |
> | 2 backtrack    | 87.2%                       |
> | 3 backtrack    | 82.6%                       |
>
> ---
>
> ### **Countdown**
> | **Data**      | **Evaluation Acc (%)** |
> |-----------------|:---------------------------:|
> | 0 backtrack    | 37.6%                       |
> | **1 backtrack**| **63.0%**                   |
> | 2 backtrack     | 48.2%                       |
> | 3 backtrack     | 44.4%                       |
>
> ---
>
> ### **Sudoku**
> | **Data**                | **Evaluation Acc (%)** |
> |---------------------|:------------------------:|
> | 0 backtrack          | 9.6%                        |
> | 1 backtrack          | 23.4%                       |
> | 5 backtrack          | 25.2%                       |
> | **10 backtrack**   | **26.9%**            |

---

> ### Author Response · Authors · 2025-11-29
> **Author Responses to Reviewer 2uuo (2/n)**
>
> > W3: The general idea of combining SFT warm-up with RL is well-trodden; the specific contribution—varying backtracking depth—is interesting but may be viewed as an empirical refinement rather than a conceptual leap.
>
> Thank you for raising this point. While SFT warm-up followed by RL is indeed a common recipe, our contribution goes beyond this pattern: we identify backtracking depth as a previously unformalized control variable that shapes RL’s exploration dynamics. Prior work has noted the existence of warm-up and backtracking phases, but none has quantified how much backtracking is beneficial, how it interacts with reward shaping, or where diminishing returns and failure modes arise. In other words, our contribution is identifying that backtracking is not a binary capability (backtrack vs. don't backtrack), but a continuous spectrum of error correction that requires hierarchical supervision. Our work provides the first systematic framework for controlling, synthesizing, and evaluating backtracking depth across tasks and model families, yielding actionable principles for RLVR training.
>
> To our knowledge, prior work that discusses backtracking does not provide the specific quantitative insights presented here, nor does it treat backtracking depth as a controllable variable with measurable effects on RL training dynamics. We view our findings as a conceptual step toward understanding and shaping robust RL-driven reasoning, rather than merely tuning another hyperparameter. We will clarify this distinction in the revised manuscript. Please let us know if there are additional aspects that would benefit from further clarification.

---

### Official Review · Reviewer_ZT8P · 2025-11-01

**Soundness:** 3
**Presentation:** 3
**Contribution:** 3
**Rating:** 6
**Confidence:** 4

**Summary:**

This work systematically analyzes the interplay between supervised fine-tuning (SFT) and reinforcement learning (RL) in enhancing reasoning abilities of LLMs, showing that short chain-of-thought (CoT) sequences moderately benefit RL warm-starts but their impact diminishes on harder tasks. By constructing synthetic datasets with controlled backtracking steps, experiments reveal that longer CoTs with backtracks improve RL training stability and effectiveness, and more challenging tasks require more backtracking during SFT. Results further show that RL training is largely insensitive to the correctness of long CoTs, focusing instead on structural patterns, providing practical guidance for optimizing LLM reasoning training strategies.

**Strengths:**

1. Very Good Research Question: How SFT & RL affects LLM reasoning
2. Good Experiment settings on SFT types / benchmarks etc.
3. Propose a practical method to boost reasoning

**Weaknesses:**

1. **Insufficient Experiment on model choices: Only Qwen-2.5-3B**. [1] have shown that Qwen has something specific when considering reasoning abilities, so I believe more **experiments on other series open-source models and sizes (maybe ~7B)** should be added.

[1] Spurious Rewards: Rethinking Training Signals in RLVR. https://arxiv.org/abs/2506.10947

**Questions:**

see weakness

---

> ### Author Response · Authors · 2025-11-29
> **Author Responses to Reviewer ZT8P**
>
> We thank the reviewer for the thoughtful assessment of our work. We appreciate the recognition of the importance of our research question,  the strength of our experimental design, and the practicality of our proposed method for improving reasoning performance. We address the reviewer’s specific concerns below:
>
> ---
>
> > W1: Insufficient Experiment on model choices
>
> Thank you for the constructive feedback. We agree that Qwen models exhibit several family-specific behaviors—particularly in reasoning-heavy settings as highlighted by [1]—and therefore should not be evaluated in isolation. To address this, we have added a second model family, Llama-3.2-Instruct-3B, and replicated our experimental setup. Despite the architectural and pretraining differences, the Llama models show the same qualitative trends, reinforcing that our findings are not Qwen-specific.
>
> Due to computational constraints, we are unable to include ~7B models in this revision, but we are actively working toward these larger-scale trainings and evaluations, and will incorporate them in a future version.
>
> Below we include the new Llama-3.2-3B results trained on our synthetic dataset, which further validate the generality of our conclusion that more difficult tasks require data with higher frequency of backtrack during the SFT stage.
>
> ### **Arc 1D**
> | **Data**     | **Evaluation Acc (%)** |
> |----------------|:------------------------:|
> |**0 backtrack**| **91.5%**                  |
> | 1 backtrack    | 85.3%                       |
> | 2 backtrack    | 87.2%                       |
> | 3 backtrack    | 82.6%                       |
>
> ---
>
> ### **Countdown**
> | **Data**      | **Evaluation Acc (%)** |
> |-----------------|:---------------------------:|
> | 0 backtrack    | 37.6%                       |
> | **1 backtrack**| **63.0%**                   |
> | 2 backtrack     | 48.2%                       |
> | 3 backtrack     | 44.4%                       |
>
> ---
>
> ### **Sudoku**
> | **Data**                | **Evaluation Acc (%)** |
> |---------------------|:------------------------:|
> | 0 backtrack          | 9.6%                        |
> | 1 backtrack          | 23.4%                       |
> | 5 backtrack          | 25.2%                       |
> | **10 backtrack**   | **26.9%**            |

---

### Comment · Area_Chair_me4a · 2025-11-27
**Reminder to authors: discussion period is about to end**

Dear authors,

This is a friendly reminder that you have until Dec 03 to respond to the initial reviews. While a response is optional, I encourage you to discuss the feedback with the reviewers to improve your submission. If you have any concerns regarding the reviews, please let me know.

Best,

AC

---

### Author Response · Authors · 2025-12-04
**Rebuttal Summary**

We appreciate the reviewers for their hard work in reviewing our paper. Below we summarize the responses to reviewers concerns and the reflected updates to manuscript (changes marked in red):

**Expanded Main Result Validation.**

We reproduced all Section 4 experiments using Llama3.2-3B-Instruct, addressing reviewers' concern about relying on a limited model family. The results closely match those obtained with the original model used, Qwen2.5-3B-Instruct, further confirming the generality of our central claim: **more difficult reasoning tasks require a higher frequency of backtracking in demonstration data during the SFT stage**. (Review `ZT8P`, `2uuo`, `FbXu`, `HHdh`)

---

**Strengthened Motivation for Studying Backtracking.**

We revised the introduction to more clearly articulate why backtracking warrants systematic study. We emphasize that backtracking is a universal mechanism for tasks that can be framed as tree search, as it encompasses both multi-step verification strategies and exploration of alternative partial solutions. This clarifies the need for controlled experimentation on backtracking frequency. (Reviewer `FbXu`: W1, Q8)

---

**Terminology Cleanup and Length-Related Clarifications.**

We updated terminology throughout the paper, replacing “short CoT’’ with self-sampled CoT and “cold-start RL’’ with pure RL. We also added token-count comparisons in Appendix E showing that self-sampled trajectories are inherently shorter than those produced by models trained on synthetic backtracking datasets. This removes ambiguity around “short CoT’’ and emphasizes that response length is not a controlled variable, but naturally increases as backtracking frequency increases. (Reviewer `FbXu`: W3, W5, W6, Q1, Q3, Q5, Q7)

---

**Revised Analysis of Sudoku Failure Modes under Pure RL.**

We expanded Section 3 to discuss why cold-start RL strategy struggles on Sudoku: the task is sufficiently hard that we couldn’t sample correct trajectories, making the subsequent RL run impossible. This highlights a fundamental limitation of Cold-start RL on certain tasks, better motivating Section 4and synthetic backtracking data. (Reviewer `FbXu`: W4, Q6, Q2)

---

**Improved Discussion of Pure RL Trajectories.**

We refined Section 3 to show that trajectories self-sampled from pure RL'ed models often lack coherent structure and fail to identify specific points of error. In contrast, synthetic backtracking datasets in Section 4 explicitly correct both issues by enforcing structured error localization and recovery. (Reviewer `FbXu`: W8, Q5)

We greatly appreciate each reviewer's comment to help us improving the quality of our work. We also thank the AC's effort in reviewing.

---

### Meta-Review · Area_Chair_3bp4 · 2025-12-24

**Summary:**

This paper conducts an investigation about the interplay between SFT and RL in improving LLM’s reasoning capabilities. It further proposes a method that adopts the optimal number of backtracking steps for effective and stable training.

Three reviewers gave the rating of 4, and one gave 6. The authors have provided detailed responses to their comments. By reading through the paper, reviewers and rebuttal, AC found some of the concerns could be solved, while leaving two critical ones unaddressed:

1.	Three reviewers asked for evaluations over larger models. Unfortunately, the authors only added the evaluation of the same 3B model. Such size is too small. It is still not sure whether the findings and method can be used for larger models, which are more common now.
2.	Three reviewers also doubted about the selection of tasks. The authors’ responses are not satisfactory. We all expect to see the extended evaluation results over more complex tasks, but the authors claimed that those may be beyond the scope of this paper, and not suitable for the study of backtracking. This is somehow indicating the limited scope of this paper.

Given the above two points, AC recommended rejection for this paper.

**Reviewer Concerns:**

Some technical clairifications have been addressed by the rebuttal, but the extension to larger models and more tasks are outstanding.

**Reviewer Scores:**

Given the unaddressed concerns shared by the reviewers, AC thinks they will not adjust their score.

---

### Decision · Program_Chairs · 2026-01-26

Reject